# The Role of Microbial Exopolysaccharides in Preventing and Treating Cardiovascular Diseases

**DOI:** 10.3390/microorganisms13071522

**Published:** 2025-06-29

**Authors:** Enrique A. Sanhueza-Carrera, Cassiopeia Cantero-Ramírez, Angel D. Montijo-Valdés, Cinthya G. Rodríguez-Valladares, Cynthia Fernández-Lainez, Itzhel García-Torres, Pedro Gutiérrez-Castrellón, José F. González-Zamora, Gabriel López-Velázquez

**Affiliations:** 1Laboratorio de Biomoléculas y Salud Infantil, Instituto Nacional de Pediatría, Secretaría de Salud, Mexico City 04530, Mexico; enriquesanhueza@comunidad.unam.mx (E.A.S.-C.); canteroramz@gmail.com (C.C.-R.); amontijovaldes@gmail.com (A.D.M.-V.); cinthyavalladares96@ciencias.unam.mx (C.G.R.-V.); garcia.itzhel@gmail.com (I.G.-T.); 2Posgrado en Ciencias Biológicas, Universidad Nacional Autónoma de México, Mexico City 04510, Mexico; 3Laboratorio de Errores Innatos del Metabolismo y Tamiz, Instituto Nacional de Pediatría, Mexico City 04530, Mexico; cfernandezl@pediatria.gob.mx; 4International Scientific Council for Probiotics, A.C., Mexico City 14340, Mexico; pedro.gutierrez@councilforprobiotics.org; 5Translational Research Center, Instituto Nacional de Pediatría, Mexico City 04530, Mexico; jf.gonzalezzamora@gmail.com

**Keywords:** exopolysaccharides, cardiovascular diseases, postbiotics, probiotics, prebiotics

## Abstract

Cardiovascular diseases (CVDs) have become one of the major global health crises of the last century, causing millions of deaths each year, and are the leading cause of disability worldwide. The pharmacological management of these conditions demands new alternative or complementary therapies due to the multiple long-term side effects experienced by patients. In this context, exopolysaccharides (EPSs) have emerged as a promising alternative due to their numerous functional properties and favorable biotechnological and medical applications for health. This review provides an overview of the properties of EPSs as bioactive agents in cardiovascular diseases, highlighting the cellular signaling mechanisms in their role as cardioprotective agents, with a primary focus on their roles as antioxidants, antihypertensives, and cholesterol regulators, and their regenerative effects on vascular epithelia, positioning EPSs as promising biomolecules for CVD prevention.

## 1. Introduction

Cardiovascular diseases (CVDs) encompass a range of disorders that affect the heart and circulatory system, resulting in various heart conditions, such as coronary artery disease, cerebrovascular disease, peripheral artery disease, rheumatic heart disease, venous thrombosis, pulmonary embolism, and stroke [1]. Recent evidence from the last Global Burden of Disease study establishes that CVDs, including stroke and ischemic heart disease, are the main causes of global mortality and a major cause of disability [2,3]. The increasing incidence and prevalence of CVDs cause 17.9 million deaths each year, representing 31% of all deaths globally. Current forecasts have determined CVDs as the global pandemic of this century, which will be responsible for 23.6 million deaths by 2030. For the Americas region, it is estimated that around 2 million people died from CVDs in 2019 [4].

CVDs have multiple and different causes, from congenital and even environmental conditions, such as air pollution, to social determinants of health, such as socio-demographic, cultural, and economic factors that impact people’s lifestyles. Lifestyle factors such as smoking, alcohol consumption, and an unhealthy diet based on processed foods with high contents of sugars, sodium, and fat, additionally joined with a lack of physical activity, are determining factors for the development of CVDs. As a result, individuals develop a pathological condition known as metabolic syndrome that includes the development of hypertension, hyperglycemia, hyperlipidemia, and insulin resistance, among other consequences [1,2,3,4,5,6,7].

Oxidative and nitrosative stresses are physiological conditions that commonly produce reactive oxygen species (ROS) and reactive nitrogen species (NOS), respectively. In metabolic syndrome, these mechanisms are exacerbated and dysregulated due to systemic inflammation, leading to multiple pathological processes, including lipid peroxidation, protein damage, DNA damage, cellular machinery dysfunction, and ultimately, loss of tissue function and structure [8]. For instance, oxidative and nitrosative stresses play a critical role in the pathogenesis of atherosclerosis, where the oxidation of low-density lipoproteins, commonly referred to as LDL cholesterol, triggers their capture by macrophages, ultimately leading to endothelial damage. ROS and NOS overproduction in metabolic syndrome are also critical factors during the development of cardiac and ventricular hypertrophy, adverse angiogenesis, and cardiac remodeling and ultimately lead to cell senescence and heart failure [6,7,8,9,10]. 

Understanding these inflammatory and oxidative processes and their role in hypertension, structural damage, and deficient angiogenesis provides new opportunities to develop therapeutic strategies for preventing and treating the biological factors implicated in metabolic syndrome that promote the development of cardiovascular diseases (CVDs) [6,7,8]. Moreover, it is important to note that there are already available drugs to cope with these pathologies. Despite the availability of effective pharmacological treatments, their widespread use is often limited by significant long-term side effects [10]. For example, beta-blockers can cause fatigue, dizziness, and erectile dysfunction in some patients, angiotensin-converting enzyme inhibitors (ACEIs) can cause persistent cough and increased blood potassium levels, and angiotensin II receptor (ARB II) antagonists may be associated with dizziness and inefficient kidney function [11,12,13].

The high number of side effects creates an urgent need to search for new alternatives based on biomolecules with antioxidant, immunomodulatory, and antihypertensive activities to prevent and treat CVDs, avoiding the adverse effects of the traditional drugs. In this context, exopolysaccharides (EPSs) are natural biomolecules that have recently gained significant relevance due to their multiple biotechnological applications in the pharmaceutical and food industries, as well as their positive health effects in vitro and in vivo [12,13,14,15,16,17,18,19,20,21,22,23,24,25].

To conduct this review, a literature search was performed in PubMed, Scopus, and Web of Science, spanning the period from 1990 to 2024. The search strategy included the following terms: “exopolysaccharides” OR “microbial polysaccharides” AND (“bacteria” OR “fungi” OR “archaea” OR “algae”) AND (“cardiovascular diseases” OR “atherosclerosis” OR “lipid metabolism”) AND (“carbohydrates” OR “biopolymers”) AND (“probiotics” OR “microbiome modulation”) AND (“angiogenesis” OR “antioxidant” OR “cholesterol-lowering” OR “tissue regeneration” OR “antihypertensive” OR “anticoagulant”). Peer-reviewed articles published in English between 1990 and 2024 were considered. Eligible studies included experimental, clinical, and review articles focusing on the role of microbial exopolysaccharides in the prevention or treatment of cardiovascular diseases, particularly regarding their angiogenic, antioxidant, cholesterol-lowering, tissue-regenerative, antihypertensive, and anticoagulant properties.

## 2. General Characteristics of Exopolysaccharides

EPSs are high-molecular-weight saccharide biopolymers produced by multiple microorganisms, including bacteria, fungi, and algae [16,19,20,21]. EPSs are secreted into the extracellular environment as they form the basis of biofilms, allowing microorganisms to adhere to a substrate, interact with other cells, and proliferate in an internal ecosystem that counteracts the effects of the external environment. Therefore, EPSs, as raw materials of biofilms, enable microorganisms to survive in stressful environmental conditions [15,20]. EPSs typically present structural arrangements including hydroxyl groups, mannose monomers, glucose, fucose, uronic acids, sulfate groups, branches, functional groups, and the presence of β and α-type glycosidic bonds [25,26,27]. EPSs can be either homopolysaccharides (HoPSs) or heteropolysaccharides (HePSs) depending on their monomeric composition (Figure 1).

Microbial EPSs present variable and complex physicochemical compositions, enabling them to perform multiple biological functions in nature. Among the structural characteristics that determine the biological functions and even the production rate and quality of EPSs are the monomeric composition, glycosidic bonds, structural branches, functional groups, molecular weight, and conformational arrangement. This functional flexibility can be exploited by biotechnological applications in several interesting ways, e.g., for the synthesis of specific EPSs with particular or enhanced biomedical properties. For instance, specific EPSs with particular biomedical properties can be synthesized naturally or artificially, depending on the culture-growing conditions, such as the pH, temperature, salinity, carbon source, and specific characteristics of the strain [25,26,27]. Thus, the described biomedical applications of EPSs to treat CVDs encompass their antioxidant, anti-inflammatory, immunomodulatory, wound-healing, cholesterol-reducing, and antihypertensive activities [20,21,22,23,24]. Furthermore, EPSs have a polymeric nature that makes them perfect for use in the development of cell scaffolds in tissue engineering, which emulates the extracellular matrix, promoting cell differentiation in tissue regeneration assays and treatments [18].

Considering the current knowledge, the objective of this review is to explore and update the described beneficial properties of microbial EPSs in counteracting CDVs, from the cellular signaling level to the vascular system level. We especially emphasize the relationship between EPSs’ structure and function, such as their antioxidant, antihypertensive, regenerative, and cardioprotective effects. Finally, we discuss the potential of employing microbial EPSs to improve or even replace traditional treatments for cardiovascular diseases, which are among the most prevalent and deadly diseases globally.

### 2.1. Microbiome Modulation and the Impact on EPSs

The modulation of the gut microbiome has a significant influence on the production, structure, and functionality of microbial EPSs. For example, it has been shown that the intake of prebiotics such as inulin and fructooligosaccharides selectively promotes the growth of beneficial bacteria like Bifidobacteria and Lactobacilli, creating a favorable environment for the synthesis of complex and bioactive EPSs by the enteric microbiota, including immunomodulatory and anti-inflammatory properties [28,29]. In addition, in vitro studies have demonstrated that EPSs themselves can act as microbiota modulators by increasing the production of short-chain fatty acids (SCFAs), organic acids, and other biomolecules that support the proliferation of beneficial bacteria, thereby contributing to the modulation of bacterial diversity in the gut and human fecal cultures [30]. Likewise, factors such as diet, antibiotic use, or fecal microbiota transplantation can significantly alter the structure of the intestinal ecosystem, leading to the reprogramming of the microbiota diversity and, consequently, the various microbial metabolic pathways involved in EPS biosynthesis [31].

### 2.2. Antioxidant Activity of EPSs

In vitro and in vivo studies highlight the antioxidant activity of EPSs produced by bacteria, unicellular fungi, basidiomycetes, and microalgae [25]. These effects include reductive power, ferrous ion chelation, the inhibition of lipid peroxidation, and increases in the activities of glutathione peroxidase (GPX), superoxide dismutase (SOD), and catalase (CAT). EPSs can also eliminate hydroxyl radicals and superoxide anions due to the presence of hydroxyl groups in the molecule, which provide hydrogen atoms that combine with hydroxyl radicals to form water. Then, carbon-free radicals can further oxidize to form peroxyl radicals, leading to the formation of harmless molecules [26].

Sulfate groups are another structural motif that influences the antioxidant activity of EPSs. It has been established that the sulfate groups in an EPS molecule can determine various conformational structures, which, in turn, increase the antioxidant activity [32,33,34,35,36,37,38]. Such conformational changes, resulting from the electrostatic repulsion between negatively charged sulfate groups, can enhance the hydration of the EPS, making it more soluble and better able to interact with aqueous solutions, thereby exposing more reactive sites, improving accessibility for radical scavenging, as well as directly capturing ROS generated during the lipid peroxidation chain. For instance, studies on *Cyclocarya paliurus* EPSs have shown increased antioxidant activity resulting from the substitution of sulfate groups at the C6 position [25]. When lactic acid bacteria (LAB) EPSs are hydrolyzed, active hemiacetal hydroxyl groups are secreted, which transfer electrons to free radicals and promote the conversion of these radicals to more stable molecules [39,40,41,42]. Studies with the *Lysinibacillus sphaericus* Ya6 strain have demonstrated that the glycosidic bonds in its EPSs facilitate efficient electron or hydrogen atom transfer, leading to the scavenging of hydroxyl radicals [43]. Furthermore, several studies have demonstrated that the reductive power of EPSs is directly proportional to their concentration in the surrounding environment. This is reinforced by studies on EPSs produced by LAB, where greater EPS production leads to better inhibitory activity against superoxide anions, hydrogen peroxide, and lipid peroxidation [40,41,42,44,45,46,47,48,49,50,51].

EPSs can act as chelators of key enzymatic cofactors, such as Fe^2+^ or Cu^2+^ ions, which are involved in the generation of ROS. The chelating properties of EPSs against transition metal ion catalysis decrease the production of hydroxyl radicals [27,52,53,54,55,56,57,58]. The antioxidant effect of EPSs can also act through cellular signaling pathways. For example, high-molecular-weight EPSs interact with Toll-like receptors (TLRs), leading to antioxidant and immunomodulatory responses, while low-molecular-weight EPSs, typically smaller than 10^4^ Da, can diffuse into the cell, potentially activating transcription factors at the nuclear level [59]. Studies have shown that EPSs isolated from *Lactobacillus rhamnosus* GG can effectively protect Intestinal Porcine Epithelial Cell line-J2 (IPEC-J2) cells from H_2_O_2_-induced oxidative stress by enhancing the Nuclear factor erythroid 2-related factor 2/Kelch-like ECH-associated protein 1 (NRF2/KEAP1) signaling pathway, the intracellular antioxidant defense regulator [60,61].

The NRF2/KEAP1 signaling pathway plays a crucial role in regulating the antioxidant response. First, under normal physiological conditions, the NRF2 protein is found in low amounts in the cytoplasm. However, when cells experience oxidative stress, the cysteine residues of KEAP1 that contain a thiol group are modified by redox reactions, leading to a change in KEAP1’s ability to degrade NRF2. Before the proteasomes degrade NRF2, it is polyubiquitinated by KEAP1 through the ubiquitin ligase complex composed of RING-box protein 1 (RBX1) and Cullin 3 (CUL3). This process increases the levels of NRF2, causing it to translocate and accumulate in the cell nucleus, where it forms a heterodimer with the sMAf protein, which recognizes the antioxidant response elements (AREs). ARE sequences are found in regulatory regions adjacent to genes involved in the antioxidant and detoxification response, such as SOD, GSH, CAT, and GPX. This initiates the cellular antioxidant response (Figure 2) [60,61,62,63,64,65].

EPSs from *Lactobacillus* sp. have been shown to decrease lipopolysaccharide (LPS)-induced inflammation by blocking ROS production through the inhibition of inducible nitric oxide synthase (iNOS) at both the mRNA and protein levels [60,61]. The *Paenibacillus polymyxa* JB115 strain synthesizes β-glucan EPSs that exhibit antioxidant properties by increasing the nitric oxide (NO) production in macrophages through the MAPK and Nuclear factor kappa-light-chain-enhancer of activated B cells (NF-κB) signaling pathways. *P. polymixa* EPSs also increased the GSH (glutathione) levels inside the liver, demonstrating an enhanced antioxidant effect [64,66,67,68]. 

### 2.3. EPS Cardiovascular Benefits in Diabetic Animal Models

EPSs have exhibited properties that counteract the pathological cardiovascular conditions associated with diabetes. A study conducted using a diabetic mouse model demonstrated that EPS administration improved the electrocardiogram pattern and inhibited the cardiotoxic effect induced by cyclophosphamide. Cyclophosphamide is a cardiotoxic agent that causes endothelial damage and cardiomyocyte destruction, leading to oxidative stress in cardiac tissue. However, the treatment with EPSs restored the attenuated QRS waves in the electrocardiogram. The study also demonstrated that EPS administration reduced the level of malondialdehyde (MDA) induced by streptozotocin in the hearts of mice. MDA is an indicator of lipid peroxidation that produces free radicals in proteins and lipids, affecting the biological properties of biomolecules [69,70,71]. These results support the ability of EPSs to maintain the heart’s structural integrity, minimize cardiac damage, and improve cardiac function, even in the presence of diabetes, likely by enhancing lipid and glucose metabolism [72].

Interestingly, the positive effects of EPSs have been demonstrated even in a nematode model. *Caenorhabditis elegans* expresses DAF-16, a transcription factor related to oxidative stress tolerance. The administration of fructan-type EPSs was able to induce the DAF-16 signaling pathway, activating the endogenous antioxidant defense system [69,73].

## 3. Antihypertensive Activity of EPSs

Hypertension is a chronic non-communicable disease affecting approximately 1.28 billion people worldwide, making it a significant risk factor for the development of pathologies such as coronary artery disease, peripheral artery disease, venous thrombosis, pulmonary embolism, and renal diseases [74,75]. Hypertension is characterized by an increase in prothrombotic activity and a reduction in fibrinolysis (blood clot lysis), causing damage mainly to the heart, kidney, central nervous system, and vascular endothelium, leading to the dysfunction of the inner layer of blood vessels. This promotes the development of atherosclerosis, which reduces the blood supply to the heart, increasing the risk of acute myocardial infarction and stroke [17,70].

The primary treatments for hypertension involve diuretic drugs, which aid in the elimination of excess water to lower blood pressure, along with ACE inhibitors, ARB II inhibitors, and calcium channel blockers. Research on hypertension has primarily focused on identifying potential inhibitors of ACEs. These enzymes are involved in two key processes that increase vasoconstriction in the system. First, ACEs promote the catalysis of angiotensin I to angiotensin II. Additionally, they promote the degradation of the vasodilator bradykinin, resulting in a significant increase in vasoconstriction. Therefore, ACE inhibitors act as antihypertensives [17,70,76]. Several studies have already demonstrated the role of EPSs as hypotensive agents. For example, EPSs produced by *Streptococcus thermophilus* and *Lactobacillus bulgaricus* isolated from “labané” showed 78.3% and 77.1% ACE inhibition, respectively, at 5 mg/mL, exhibiting a dose-dependent effect [77]. Similarly, EPSs produced by *L. plantarum* 1.0665, with a low molecular weight composed mainly of mannose, showed 38.10% inhibition of ACEs [78]. The polysaccharides produced by *Pleurotus eryngii* were evaluated for their ACE inhibitory activity when fermented by *Streptococcus thermophilus* in non-fat milk. The results showed maximum ACE inhibitory activity on day 14 [71,79,80,81].

### Antihypertensive Effects of EPSs of Probiotics

The *Lactobacillus casei* LC2W strain, isolated from skim milk fermentation, synthesizes an EPS identified as LPC1. Findings indicate that LCP1 decreased the systolic blood pressure in hypertensive rats on the first day, with a maximum reduction observed 2 to 4 h after oral administration. However, the effect on blood pressure was not observed after 24 h. Subsequently, the effect was observed in rats fed with LCP1 for 6 days in a dose-dependent manner [82]. Another study evaluated the antihypertensive role of EPSs produced by kefir. Kefiran-type EPSs produced by the strain *Lactobacillus kefiranofaciens* WT-2BT, composed of glucose and galactose, showed a dose-dependent hypotensive effect when administered to a hypertensive rat model. Biochemical analyses of sera and aortic arteries from the rats indicated significantly lower ACE inhibition [82]. *L. plantarum* KX881772 and KX881779, isolated from camel milk used for Akawi cheese production, showed significant ACE inhibitory activity when their EPSs were incorporated into cheese, demonstrating a higher inhibitory effect compared to that of the controls [83]. EPSs produced by *Streptococcus thermophilus* 1275 (both capsular and ropy variants) during yogurt production also showed significant enzymatic inhibition of ACEs, particularly in yogurt combined with EPSs and inulin [84]. Another example of *L. plantarum* is the JLK0142 strain, which was supplemented in low-fat cheddar cheese production for 90 days, showing the highest ACE inhibition rate compared to that of the controls after 90 days [85]. The *Lachnum* sp. YM130 strain synthesizes benzoylated and acetylated EPSs, designated as ALEP-2 and BLEP-2, respectively. These EPSs demonstrated cardioprotective effects in a diabetic mouse model, characterized by increased NO production and decreased endothelin-1 levels, which promote a balance between vasoconstriction and vasodilation, thereby exerting a protective effect on the myocardium [86].

These findings suggest that the structural complexity and high molecular weight of EPSs may not necessarily play a favorable role in ACE inhibition, indicating that EPS functionality primarily depends on specific structural motifs and even the culture conditions. For instance, EPSs exhibiting benzoylation and acetylation motifs, such as those synthesized by the *Lachnum* sp. YM130 strain (as previously explained), have demonstrated significant cardioprotective effects. The potential role of EPS acetylation in counteracting hypertension can be elucidated by comparing it with the acetylation mechanisms observed in other pharmacological agents. For example, aspirin’s acetylation ability allows it to irreversibly inactivate enzymes such as Cyclooxygenase-1 (COX-1) and Cyclooxygenase-2 (COX-2). While primarily recognized for its antiplatelet and anti-inflammatory properties, aspirin, particularly at low doses, has been shown to modestly influence blood pressure by affecting the synthesis of prostaglandins and thromboxanes, which play crucial roles in regulating vascular tone [87].

Further studies are needed to propose a possible mechanism for how EPSs can inhibit ACE activity. Testing EPSs with variable molecular weights and diverse monomeric compositions may elucidate a potential mechanism of inhibition against this enzyme.

## 4. Cardiovascular Regeneration Properties of Exopolysaccharides

Angiogenesis promotes tissue regeneration by growing new blood vessels from existing ones [88]. This process involves the development, repair, and formation of tissues through the proliferation, migration, and differentiation of endothelial cells [32,33]. Therefore, angiogenesis is fundamental for cardiovascular health, as it allows for the survival, integrity, and proper functioning of cardiomyocytes by providing nutrients and adequate oxygenation to every cell of the body [34]. It is essential to consider that in myocardial infarction, the damage to cardiomyocytes can be irreversible due to the processes of ischemia and necrosis that occur. Finally, this cardiac damage makes the heart more vulnerable to other chronic diseases, increasing the chances of death [34,35].

Along this line, EPSs have shown myocardial-regenerative effects, even amidst the chronic damage provoked by CVDs. EPSs mitigate this damage preventively by promoting several differentiation, regenerative, cell migration, and neovascularization signaling pathways, thereby favoring the differentiation and migration of resident cells towards damaged areas, which can contribute to the partial functional regeneration of myocardial tissue [89,90].

EPSs of multiple natures and origins can exhibit pro-angiogenic activity, enhancing the regeneration process through cellular signaling pathways. Furthermore, EPSs can be used as scaffolds to promote cell differentiation in tissue engineering, structurally improving cardiac tissue regeneration. This finding emphasizes the potential of EPSs as valuable biomolecules in prophylactic and therapeutic treatments for cardiovascular diseases. Glucan-type EPSs have also been shown to have effects on angiogenesis and re-epithelialization. For instance, the *Bordetella* sp. B4 strain synthesizes glucan-type EPSs that exhibit protective effects on umbilical vein endothelial cells (HUVECs) against high glucose levels during angiogenesis [36].

Collectively, these types of EPSs induce the nuclear export of Histone deacetylase 5 (HDAC5) in HUVECs, triggering the activation of the myocyte enhancer factor 2 (MEF2)-dependent gene regulation pathway. This favors endothelial cell migration and neovascularization [37]. Additionally, it has been determined that β-glucans phosphorylate serine/threonine-type protein kinase (Akt) and ERK1/2 factors in endothelial cells, inducing angiogenesis [38,91].

The fucoidan-type polysaccharides of brown algae cell walls have been characterized by studies in the generation of pro-angiogenic factors and their interaction with extracellular matrix components to stimulate the re-epithelialization process [39]. There are microorganisms that naturally produce fucoidans as EPSs, which can be derivatized to increase their degree of sulfation, as we have mentioned so far [40,41,42]. Such was the case of the study carried out by Matou and collaborators, in which they obtained a sulfated EPS from *Alteromonas infernus*. This type of EPS increased the proliferation of human umbilical vein endothelial cells (HUVECs) by 52% after 3 days of treatment with FGF-2, whereas it increased by 87% after 6 days of treatment with VEGF. They also confirmed that cell migration increased by 28%. Additionally, it enhanced the density of the capillary network in HUVECs by 56% and, in conjunction with FGF-2, facilitated the formation of closed and defined vascular tubes. Additionally, the studied EPSs promoted the formation of vascular tubes by increasing the density of this network by 3.8 times [40]. Therefore, it was concluded that this sulfated EPS promotes in vitro angiogenesis through cell proliferation and the formation of vascular tubes.

HE800 is an EPS of high molecular mass (800 kDa) from the marine extremophile *Vibrio diabolicus* (Hyalurift^®^), also studied for its properties in angiogenesis and re-epithelialization. HE800 shows linear physicochemical conformation with tetrasaccharide repeats that include glucuronic acid and hexosamine of the glycosaminoglycan type [41]. Various studies with this EPS have established its effect on angiogenesis and re-epithelialization, demonstrating that its physicochemical conformation closely resembles the extracellular matrix components of epithelial cells, such as collagen. Results in a rat model showed that HE800 contributes to promoting the structuring of the extracellular matrix after 15 days of treatment, with tissue regeneration and neovascularization around the lesioned bone tissue [44]. The authors concluded that this bacterial EPS is a type of hyaluronic acid-like EPS that stimulates several growth factors, promoting regeneration and neovascularization. Results also demonstrate its ability to interact with divalent cations, such as calcium, which contributes to the firmness of the newly forming tissue and protects against damage caused by enzymes that interfere with re-epithelialization. Then, its application stimulates fibroblasts to generate collagen in the extracellular matrix [42].

As in other studies, this EPS was also sulfated, yielding derivatives of low molecular weight called HE800 DROS, which exhibit physicochemical similarities to heparan sulfate. This EPS was able to stimulate fibroblast proliferation by 50% and inhibit the production of extracellular matrix metalloproteinases. The mechanism of these effects is based on the inhibition of the stimulation produced by IL-1β, both from the free gelatinase A enzyme and stromelysin-1. It prevents these proteolytic enzymes from degrading components of the extracellular matrix, including proteoglycans, fibronectin, type IV collagen, and other components involved in inflammatory and tissue remodeling processes [41]. Altogether, this favors the generation of tropocollagen and the cross-linking of fibrillar formation in the extracellular matrix. In the context of cardiovascular health, this inhibition is essential, as it helps minimize structural damage to the heart and blood vessels during inflammatory or ischemic events. A more supportive environment for tissue regeneration is established by maintaining the integrity of the extracellular matrix. Inhibiting matrix metalloproteinases (MMPs) promotes cellular repair, revascularization, and the functional restoration of the affected tissues. Table 1 summarizes some key features of EPSs that may contribute to cardiovascular health.

### 4.1. EPS Nanofibers in Cardiovascular and Tissue Regeneration

EPSs have a polymeric nature that makes them perfect candidates for use in the development of cell scaffolds in tissue engineering, which emulates the extracellular matrix, promoting cell differentiation in tissue regeneration, assays, and treatments. For instance, EPSs of *Lasiodiplodia theobroma* were applied as a hydrogel on Wistar rats, where they stimulated collagen fiber formation, cell proliferation, and re-epithelialization [45].

Dextran is a type of EPS consisting of repeated D-glucose units linked by α-1→6 glycosidic bonds and is produced by *Streptococcus* sp. and *Leuconostoc* sp. bacteria. Pullulan is an EPS formed by α-(1→4) and α-(1→6)-glucan maltotriose and is produced by the fungus *Aureobasidium pullulans.* Oves et al. demonstrated that dextran and pullulan EPSs can be used to develop nanofibers with mechanical properties similar to those of human arteries, with an average diameter of 323 nm [46].

The study conducted by Shi et al. also employed pullulan and dextran EPSs to synthesize nanofibers that promoted endothelial cell adherence, stability, and proliferation over a 14-day period. The nanofibers promoted smooth-muscle alpha-actin expression, thereby stimulating the formation of a contractile cell phenotype. Furthermore, the nanofibers maintained the stable expression of von Willebrand factor (vW) and Platelet Endothelial Cell Adhesion Molecule-1 (CD31) markers, promoting endothelial cell adhesion [47].

Moreover, EPS nanofibers were used to generate porous scaffold fibrils with pore sizes of 42 μm. These scaffolds demonstrated significant cell proliferation, as they allowed the correct infiltration of HUVECs into the pores from the first day of culture, remaining stable and metabolically active throughout the 7-day trial. Additionally, this cellular model also expressed CD31 and von Willebrand (vW) markers, indicating angiogenesis [48,49].

### 4.2. EPSs in Other Tissues: Regeneration Approaches

Madeiros et al. evaluated the ability of the *Saccharomyces cerevisiae* β-glucan-type EPS to promote ulcer cicatrization, an EPS characterized as structurally similar to the polysaccharides of brown algae, *Laminaria* sp. The study was performed on male and female patients aged from 42 to 75 years with long-standing ulcers resulting from chronic venous insufficiency. The topical application of this EPS on the ulcers led to the proliferation of fibroblasts and the synthesis of collagen fibers, indicating wound healing. Finally, new epithelium formation, vascularization, and angiogenesis were observed in all patients treated with this EPS [50].

EPSs from *Aureobasidium* sp. were effective at treating wounds in a diabetic mouse model. The treatments enhanced angiogenesis and re-epithelialization, as well as the differentiation of microvessels and fibroblasts, which is mediated by TGF-β1, by reducing the presence of polymorphonuclear inflammatory cells, macrophages, and lymphocytes [51]. The supernatant obtained from the *L. acidophilus* ATCC 4356 and 43121 strains was injected into rodents, inducing the proliferation of large blood vessels, promoting wound healing, and reducing inflammatory cellular infiltrate [109].

The chemically complex EPS of heterotrophic bacteria isolated from the fungus *Sarcodon aspratus* is composed of glucose, rhamnose, mannose, galactose, glucosamine, and a small amount of fucose, fructose, and galacturonic acid. Studies have demonstrated the capacity of the EPS to stimulate the proliferation, migration, and tube formation of HUVECs in rabbits. These properties are explained by the MAPK signaling pathway activation, the increased expression of p21 protein, the intercellular adhesion molecule 1 (ICAM1) phosphorylation of p38, and c-Jun N-terminal protein kinase (JNK) (Figure 3) [107].

In another study, the properties of EPSs isolated from Tibetan kefir fermentation, using whey as the medium, were evaluated in vitro. This EPS is composed of galactose, glucose, mannose, and very small concentrations of rhamnose and arabinose. This EPS showed a pro-angiogenic effect at concentrations of 192 ng/mL, which caused a 160% increase in new vessel formation, very similar to the effect produced by VEGF (170%), and also showed anti-inflammatory properties by inhibiting the hyaluronidase enzyme with a minimum activity of 2.08 mg/mL of EPSs and a maximum activity of 2.57 mg/mL [110].

Based on the current evidence described, we can elucidate some of the molecular mechanisms by which glucan-type EPSs, sulfated EPSs, and other EPSs interact with multiple signaling pathways, leading to health benefits. First, once glucan-type EPSs encounter tyrosine kinase receptors, such as VEGFR and FGFR, they subsequently activate PI3K, Akt, and mTOR proteins through phosphorylation. This leads to the expression of genes involved in cell proliferation and angiogenesis, including KLF2, ERG2, NR4A2, and MEF2. Furthermore, mTOR can also phosphorylate HDAC5, which, under conventional conditions, acts as a repressor of MEF2; however, once phosphorylated, it allows for the nuclear translocation of MEF2, enabling the transcription of pro-angiogenic genes. Additionally, glucan-type EPSs promote the activation of the RAS signaling pathway by the SOS adaptor protein, which phosphorylates RAS and, subsequently, MEK and ERK 1/2. The latter travels to the nucleus to activate the transcription factors MEF2, KLF2, and Elk-1, which modulate the expression of genes involved in regeneration, such as c-fos and c-jun, as well as the activity of other key factors, including NF-κB and the inflammatory response. Additionally, sulfated EPSs can also activate the RAS and Akt-mTOR signaling pathways, thereby enhancing the angiogenic and regenerative activity of cardiovascular tissue. Other EPSs can also stimulate cardiovascular regeneration through p38 phosphorylation and the subsequent activation of the MAPK cascade, which promotes the constriction of the aortic annulus [38,111,112] (Figure 4).

Undoubtedly, therapies based on the use of synthetic or naturally derived microbial EPSs to treat CVDs represent promising therapeutic alternatives, given that EPSs contribute to a healthier state by modulating the pathological metabolism associated with CVDs. In addition, beyond their metabolic benefits, EPSs can also serve as biomaterials for scaffold construction in CVD treatment, as they facilitate structural regeneration within tissue engineering approaches.

## 5. Anticoagulant Activity of Exopolysaccharides

The formation of blood clots impacts the development of cardiovascular diseases, especially those associated with myocardial infarction and stroke, such as coronary artery disease, peripheral arterial disease, and venous thrombosis. This occurs when a blood clot or plaque blocks the blood flow, causing ischemia in organs. Therefore, preventing the formation of clots or eliminating them with anticoagulant molecules is crucial for treating these pathologies [113,114].

Blood coagulation occurs when fibrinogen is converted into fibrin, initiating the formation of a clot or thrombus. Heparin is a sulfated glycosaminoglycan commonly used as an anticoagulant in clinical settings due to its high affinity for antithrombin III. Heparin inactivates enzymatic pathways in the coagulation cascade, including IXa, Xa, XIa, XIIa, and thrombin. Therefore, we can say that heparin presents fibrinolytic activity (Figure 3) [115,116,117,118].

Studies have demonstrated that glucan-type EPSs from the *Lactiplantibacillus plantarum* 47FE, *Lentilactobacillus pentosus* 68FE, and *Lentilactobacillus pentosus* 14F strains, isolated from Egyptian cheeses, exhibited anticoagulant properties [119]. Furthermore, dextran-type EPSs from the *Enterococcus faecalis* Esawy KR758759 strain exhibited 40% fibrinolytic (anticoagulant) activity, highlighting the role of urokinase in fibrinolytic activity [120].

Sulfated EPSs have shown high efficiency in prolonging coagulation times and lysing blood clots, acting as anticoagulants [121,122,123,124,125,126,127,128]. For instance, *Alteromonas infernus* is a bacterium isolated from hydrothermal vents that synthesizes EPSs with 10% sulfate. These extremophile EPSs demonstrated an anticoagulant mechanism by inhibiting thrombin generation in plasma, similar to heparin, but with a significantly longer effect [129]. Also, the sulfation modification of a marine strain of *Enterobacter* sp. exhibited a fibrinolytic effect, completely inhibiting fibrin formation [130]. This can occur by delaying the thromboplastin-activated thrombin generation, resulting in a significantly longer lag phase of the contact-induced thrombin generation system compared to that of heparin and fucoidan, as demonstrated for various EPS derivatives [129,131].

The *Botryosphaeria rhodian* MAMB-05 strain and *Lachnum* sp. produce sulfated EPSs that exhibit better coagulation times compared to that of heparin [132]. Other studies with *Lachnum* sp. have demonstrated the inhibition of fibrinogen synthesis in mouse livers and a reduction in the fibrinogen concentration in mouse plasma, resulting in improved coagulation times [133]. Researchers have suggested that sulfated EPSs with anticoagulant activity exhibit a significant antithrombotic effect by activating antithrombin, thereby contributing to increased heparin cofactor II activity and a reduced risk of bleeding [134].

The anticoagulant activity of sulfated EPSs depends on the high content of sulfate groups and the allosteric position where these functional groups are located. For instance, dermatan sulfate EPSs with sulfated galactosamine showed an affinity for HCII, while dermatan with sulfate at position six did not exhibit anticoagulant activity despite its high sulfate content. Furthermore, other studies have confirmed a correlation between their role as anticoagulants and the presence of sulfated fucose branches [135]. The presence of disulfate saccharide residues at specific positions is a requirement for their anticoagulant role. In fact, different structural conformations identify not only the anticoagulant effect of sulfated EPSs but also their mechanism of action [135,136].

Considering the evidence, sulfated EPSs are evidently among the primary microbial EPSs that exhibit anticoagulant effects, and their interaction with the intrinsic and extrinsic coagulation cascade signaling pathways is well understood. EPSs demonstrated an anticoagulant mechanism by inhibiting thrombin generation in plasma, in a manner similar to heparin, but with a significantly longer effect [129]. The coagulation cascade is triggered by vascular injury and initiated through two primary pathways, the intrinsic and extrinsic pathways, which converge at a common terminal phase, leading to fibrin generation and stable clot formation. The intrinsic pathway is activated by endothelial disruption, initiating a sequential proteolytic cascade with XII, XIIa, XI, XIa, IX, IXa, VIII, and VIIIa. In parallel, the extrinsic pathway is rapidly initiated by the exposure of tissue factor (TF) at the site of trauma, facilitating the direct activation of factor X. Both pathways converge at the activation of factor Xa, which catalyzes the conversion of prothrombin into thrombin. Thrombin, a central effector protease, subsequently cleaves fibrinogen into fibrin monomers that polymerize via activated factor XIII (XIIIa) to form a stable fibrin meshwork. This process is modulated by the anticoagulant protein antithrombin III, which inhibits thrombin and other key serine proteases, such as Xa and IXa. The activity of antithrombin III is significantly enhanced in the presence of heparin. Additional regulatory factors, such as heparin cofactor II (HCII), contribute to the inhibition of thrombin, ensuring spatial and temporal control of the clot formation. The interaction of sulfated EPSs with key components of this pathway could inhibit thrombin generation and thereby prevent clot formation (Figure 4) [136,137].

## 6. EPSs Improve Lipid Metabolism and Cholesterol Reduction

Cholesterol is a structural and signaling sterol that is especially abundant in the cell membranes of metazoans. It is the precursor of vitamins, sex hormones, and other steroids [107]. Since cholesterol promotes the accumulation of atheroma plaque and foam cells in the walls of arteries, excess cholesterol (hypercholesterolemia) favors the development of coronary artery disease and peripheral artery disease. It is a significant risk factor for the development of CVDs [138,139].

EPSs can effectively inhibit the activity of pancreatic lipases depending on the composition of monosaccharides, especially those with high molecular weights and monosaccharide diversity. This is because the ionic charges and sulfate groups of EPSs interact with the amino groups of lipase enzymes, inactivating them [140]. EPSs also affect the expression of hepatic enzymes involved in bile salt biosynthesis, lipid metabolism, and cholesterol biosynthesis. Experimental evidence has established that β-glucan-type EPSs increase the mRNA expression of the Cyp7a1 enzyme. This enzyme belongs to the cytochrome P450 family and catalyzes the first step of the metabolic pathway that converts cholesterol into bile acids. Treatment with β-glucan-type EPSs reduces the mRNA expression of Cd36, which is a platelet adhesion protein involved in lipid transport. Together, these changes in gene regulation affect the expression of soluble vascular cell adhesion molecule-1 (sVCAM-1), an inflammatory marker in atherosclerosis that has been decreased by the action of EPSs from LAB [141].

Ashaq et al. [142] administered kefir to rats with fatty livers and observed the decreased expression of fatty acid synthase (FASN) and peroxisome proliferator-activated receptor gamma (PPARγ), along with an increase in the expression of IL-18 at the ileum. These factors, which are involved in the inflammasome and are regulators of lipid metabolism and hepatic fat accumulation, collectively led to the reduced absorption of cholesterol at the intestinal level and its subsequent elimination through feces in animal models [143,144].

Moreover, experimental evidence indicates that bile salt production also induces the expression of glycosyltransferase enzymes involved in EPS polymerization and branching. It has been observed that under stress conditions, such as those caused by bile salts, EPSs tend to be produced in more significant quantities and adhere to the walls or membranes of bacterial cells, forming a capsule. This type of EPS provides a protective barrier against bacteria, allowing them to survive in adverse conditions. When this phenomenon of increased EPSs mediated by bile salts occurs, there is a positive correlation between the EPS production and bile salt production; thus, when both phenomena are favored, bile acids are synthesized from cholesterol at the hepatic level, directly impacting the reduction in lipids and cholesterol [145].

The high molecular weight of EPSs gives them a structure of highly hydrophilic viscous gels forming mucous hydrocolloids that adhere to cellular epithelia, as bacterial biofilms do. In the intestine, for example, by adhering to cells, these gels prevent the intestinal absorption of cholesterol, cholate, triglycerides, and other lipids. Additionally, the gels provide a physical barrier that prevents lipids from interacting [97,146,147,148,149,150,151,152,153,154,155]. The effect of EPSs also influences lipid and triglyceride metabolism. It has been established that activated TLR2 receptors induce a decrease in triglycerides. EPSs from the *L. rhamnosus* GG strain decreased adipogenesis and triglyceride accumulation by increasing the expression of TLR2 on adipocytes and fibroblasts in the adipose tissues and livers of mice. This produced an anti-obesity effect, as the size of the adipocytes decreased due to increased lipolysis, reduced visceral fat content, and decreased blood triglyceride levels [156]. EPSs from *Bacteroides fragilis* can activate TLR2, which inhibits Th17 cells and promotes Treg cell expression, reducing adipogenesis [157].

In another study, an EPS from the *Lachnum* sp. was modified with selenium, and its effect was evaluated in mice fed a high-fat diet. This resulted in the suppression of the aP2 (fatty acid-binding protein 2) and FAS (fatty acid synthase) genes involved in the transport, storage, and synthesis of triglycerides in adipocytes [133]. EPSs from *Enterobacter cloacae* Z0206 were evaluated in diabetic mice, showing the increased expression of GK (glucokinase), Glut2, Sirt1, AMPK, and pAMPK, which participate in glucose metabolism, and the decreased expression of G6P (glucose-6-phosphatase). These effects resulted in the increased expression of genes involved in β-oxidation and lipid metabolism (ATGL, HSL, and CPT1) and FAS (fatty acid synthase), improving the transport and storage of fatty acids for energy production [158].

Microorganisms capable of synthesizing EPSs can exhibit cholesterol-reducing properties that are independent of the direct effects of the EPSs themselves. For instance, in the *Lactobacillus*, *Bifidobacterium*, and *Lactococcus* genera, it has been demonstrated that they incorporate cholesterol into their cell walls. This cholesterol-binding property has even been detected in dead bacterial cells, indicating that it depends on the physicochemical characteristics of the cholesterol within the walls or membranes. The interaction between bacteria and cholesterol provides structural stability and more membrane fluidity, allowing bacteria to improve their resistance to lysis and osmotic pressure. LAB also metabolize cholesterol to utilize it as a carbon source, yielding sterols, simple fatty acids, and coprostanol for energy. Another peculiarity of LAB is that they disrupt cholesterol micelles by modifying unconjugated bile salts with glycine or taurine, thereby preventing efficient cholesterol absorption in the gut [159,160].

Studies have revealed that several LABs inhibit the Niemann-Pick C1-Like 1 (NPC1L1) cholesterol transporter in the liver through propionate synthesis and the inhibition of the HMG-CoA reductase enzyme (HMGCR), resulting in reduced cholesterol absorption [161,162,163,164,165]. Furthermore, in some bacterial strains, the metabolism and genes involved in cholesterol reduction have been elucidated, determining that the final products are precursor amino acids of the citrate cycle and pyruvate metabolism, and they also regulate the genes involved in the synthesis of hydrolase and phosphotransferase enzymes, which are related to permease transport proteins and ATP-binding proteins (ABC transporters), which together regulate sterol homeostasis and function as cholesterol transporters across cell membranes [166].

Based on previously described mechanisms in the literature, the principal signaling pathways through which EPSs exert beneficial effects on cholesterol reduction can be synthesized as follows: EPSs first interact with TLR2 on the cell membrane, triggering downstream signaling through the adaptor protein MyD88. This activates both the IKK complex and the MAPK pathway, leading to the nuclear translocation and activation of the transcription factors NF-κB and AP-1, which promote inflammatory responses. However, this signaling cascade also contributes to the suppression of the nuclear receptor PPARγ (peroxisome proliferator-activated receptor gamma). In parallel, simple fatty acids derived from EPS metabolism penetrate the cell membrane and reach the nucleus, where they activate PPARγ. This receptor, when bound to the Small Ubiquitin-Like Modifier protein (SUMO), inhibits the overactivation of NF-κB and promotes the expression of genes involved in lipid metabolism, including ATGL, HSL, and CPT1 (involved in lipid degradation). Concurrently, PPARγ activation downregulates lipogenic genes, including Fabp4, aP2 (involved in the intracellular transport of fatty acids), and FAS (involved in fatty acid synthesis) (Figure 4).

These events converge on the activation of the AMPK and pAMPK pathways, which enhance metabolic regulation. Specifically, AMPK activation leads to the overexpression of the glucose transporter GLUT2, the glucokinase enzyme (GK) (both involved in glucose uptake and phosphorylation), and the regulatory protein SIRT1 (involved in energy regulation and cellular longevity) while reducing the expression of G6P (involved in gluconeogenesis), thereby promoting glucose uptake and utilization. Simultaneously, Cyp7a1 (involved in bile acid synthesis), Cd36 (involved in fatty acid uptake), and HMG-CoA reductase (involved in cholesterol synthesis) increase their expression. Ultimately, these metabolic and anti-inflammatory shifts foster an immunomodulatory environment promoted by EPSs, characterized by the activation of M2 macrophages and regulatory T cells, while inhibiting pro-inflammatory M1 macrophages and Th17 cells (Figure 5) [167,168].

## 7. Conclusions

Considering the current knowledge, it is important to recognize and emphasize the significant health benefits that microbial EPSs can offer for the development of alternative treatments for CVDs. The medical potential of EPSs is further strengthened by their ability to exhibit diverse pharmacological properties influenced by multiple and variable factors, such as the strain itself or the culture conditions. This review supports the antihypertensive, anticoagulant, antioxidant, and regenerative properties already attributed to EPSs, as well as the presence of beneficial EPSs in probiotic foods, such as kefir, cheese, and milk, highlighting the importance of a balanced diet in preventing CDVs. Given that CVDs are among the most widespread diseases globally, it is vital to continue investing research efforts into the role of EPSs in improving healthier conditions [169,170,171,172,173].

In general, EPSs are biomolecules that offer significant therapeutic potential in the treatment and prevention of cardiovascular diseases. The ability of exopolysaccharides to modulate mechanisms such as cellular signaling, cholesterol regulation, antioxidant effects, and antihypertensive action makes them a leading candidate against conventional pharmacological therapies. Moreover, their regenerative function in vascular epithelia underscores their potential for repairing and protecting the cardiovascular system. If future research in this field progresses, EPSs could become a crucial tool in cardiovascular medicine.

Nonetheless, to achieve this goal, challenges related to standardizing methods for producing high quantities of pure and homogeneous EPS structures must be overcome. Such difficulties arise because the EPS structure and function are highly dependent on the culture conditions and the characteristics of the microorganism strain used for their production. Furthermore, clinical trials to demonstrate the efficacy of using exopolysaccharides for treating these kinds of pathologies are currently lacking. Therefore, a greater effort is required to convince healthcare decision-makers to consider exopolysaccharides as a suitable alternative for treating conditions such as CVDs.

## Figures and Tables

**Figure 1 microorganisms-13-01522-f001:**
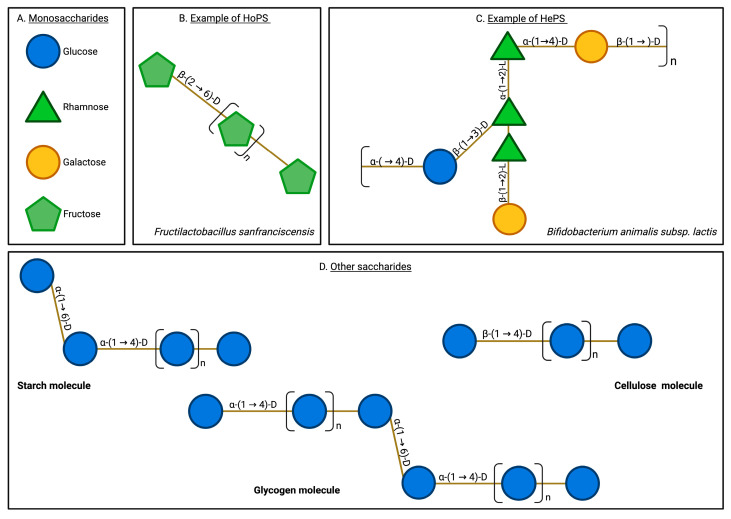
Exopolysaccharide and other saccharide structures. EPSs are characterized by their structural complexity, encompassing diverse monomeric compositions and varied glycosidic linkages. They typically include monomers such as glucose, rhamnose, galactose, and fructose, linked by various bonds, including β-(2→6), β-(1→2), and β-(1→3). In contrast, other natural saccharides, such as glycogen and cellulose, exhibit a more limited variety of monomers and bonds. Additionally, EPSs are uniquely secreted into the extracellular milieu, a characteristic not shared by other saccharides. Monosaccharides (**A**), homopolysaccharides (**B**), heteropolysaccharides (**C**), and other saccharides (**D**).

**Figure 2 microorganisms-13-01522-f002:**
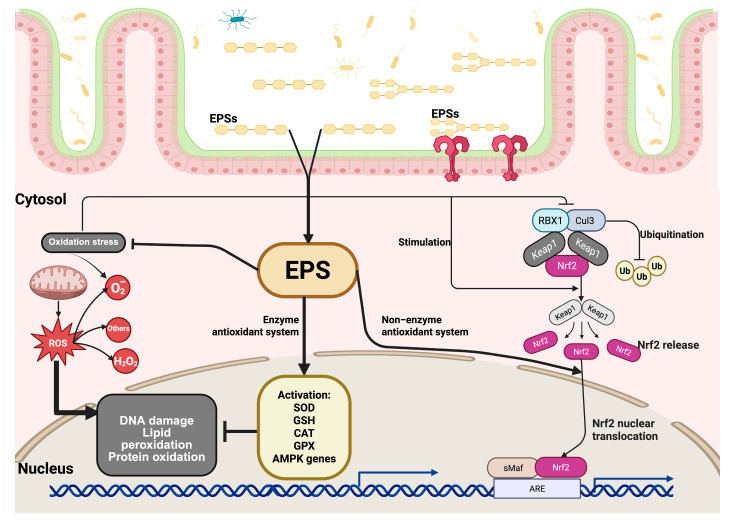
Signaling pathway of EPSs under oxidative stress. After being absorbed through the intestinal epithelium, EPSs enter the cytosol and initiate protective intracellular responses. Oxidative stress leads to cellular damage, including lipid peroxidation, protein oxidation, and DNA damage. EPSs mitigate these effects by enhancing antioxidant defenses through two major mechanisms: (1) EPSs stimulate the enzymatic antioxidant system by promoting the activation of key antioxidant enzymes, such as superoxide dismutase (SOD), glutathione (GSH), and catalase (CAT), and genes regulated by AMP-activated protein kinase (AMPK), thereby reducing ROS levels and protecting against oxidative injury. (2) EPSs activate the non-enzymatic antioxidant pathway by modulating the nuclear factor erythroid 2 (Nrf2)-related signaling cascade. Under normal conditions, Nrf2 is bound in the cytoplasm by its repressor KEAP1, which facilitates its ubiquitination and degradation through the Cullin 3 (CUL3)/RBX1 E3 ligase complex. Upon stimulation by EPSs, the interaction between Nrf2 and KEAP1 is disrupted, leading to the release and stabilization of Nrf2. The released Nrf2 is translocated to the cell nucleus, where it heterodimerizes with small Maf proteins (sMaf) and binds to antioxidant response elements (AREs) in the promoter regions of target genes, inducing the expression of cytoprotective and detoxifying genes that are involved in maintaining redox homeostasis.

**Figure 3 microorganisms-13-01522-f003:**
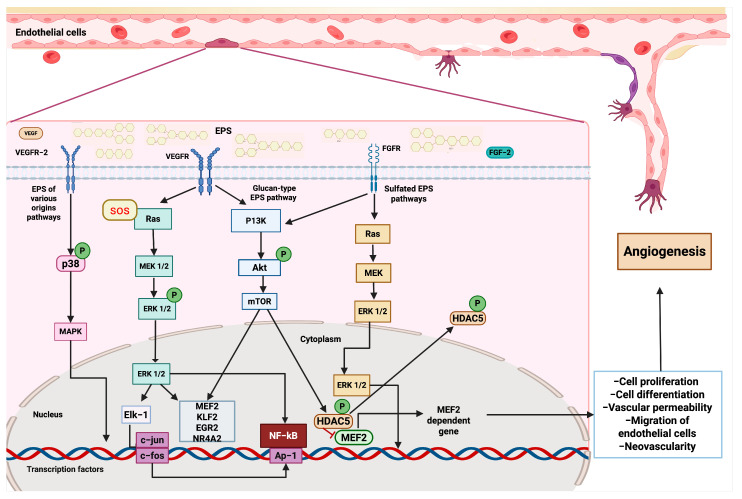
EPSs activate angiogenic pathways. After binding VEGFR (vascular endothelial growth factor receptor) and FGFR (fibroblast growth factor receptor), EPSs initiate downstream pathways that converge on the modulation of transcription factors and the expression of genes involved in endothelial cell function and vessel formation. Several EPSs can interact with VEGFR-2, triggering the activation of the SOS response, which stimulates the Ras-MEK-ERK cascade. ERK1/2 activates transcription factors, such as Elk-1, leading to the expression of immediate–early genes, including c-jun and c-fos. This activation promotes the proliferation and differentiation of endothelial cells. Parallel signaling through p38 MAPK also contributes to transcriptional activation and cell survival. Glucan-type EPSs engage VEGFR and activate the PI3K-Akt-mTOR signaling pathway. Akt phosphorylation leads to mTOR activation, which promotes cell growth and modulates the activity of NF-κB and AP-1, thereby facilitating the transcription of genes involved in angiogenesis. Sulfated EPSs interact with FGFR, initiating a Ras-dependent cascade that activates MEK and subsequently ERK1/2. In the cytoplasm, ERK1/2 phosphorylates HDAC5 (histone deacetylase 5), leading to its inactivation and allowing the transcription factor MEF2 (myocyte enhancer factor 2). MEF2 promotes the expression of specific target genes critical for endothelial cell migration, differentiation, and vascular remodeling.

**Figure 4 microorganisms-13-01522-f004:**
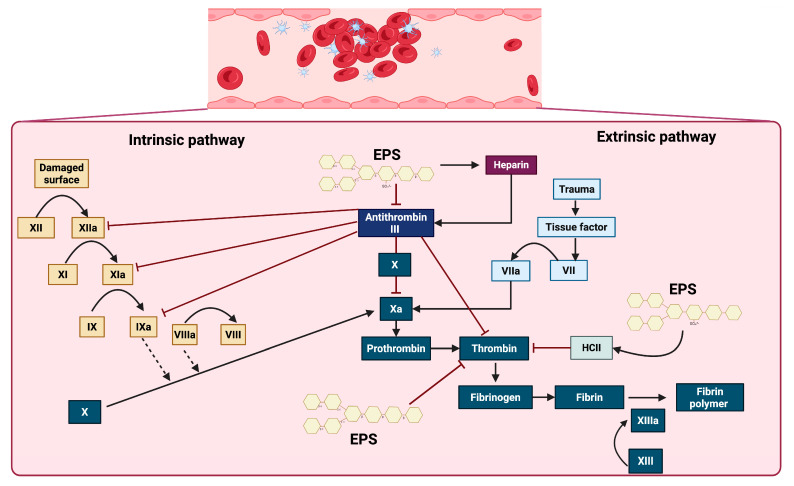
Effect of EPSs on some proteins and factors in the coagulation cascade. EPSs exert anticoagulant effects primarily through the activation of antithrombin III and heparin cofactor II (HCII), thereby inhibiting key coagulation factors, including XIIa, XIa, IXa, Xa, and thrombin. Additionally, EPSs improve the action of heparin, which further enhances the inhibitory activity of antithrombin III. These interactions collectively prevent the conversion of prothrombin to thrombin and fibrinogen to fibrin, ultimately impairing the formation of the stabilized fibrin polymer catalyzed by factor XIIIa.

**Figure 5 microorganisms-13-01522-f005:**
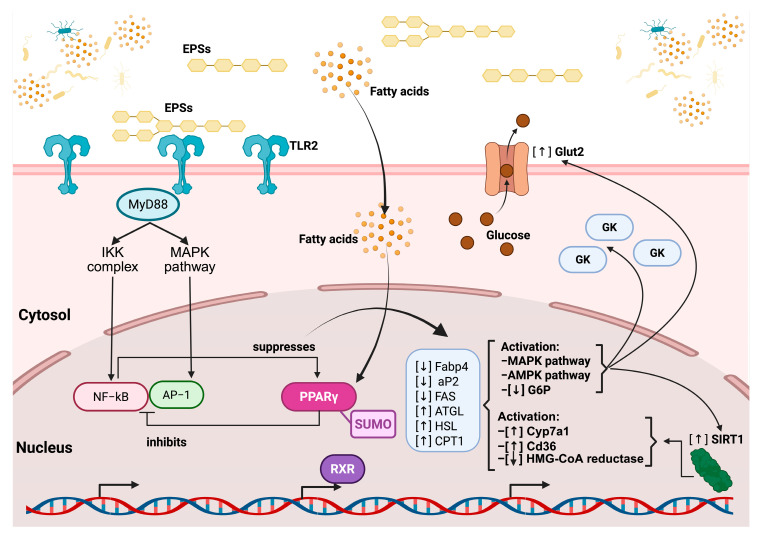
The regulatory pathway of EPS on TLR2 and the nuclear receptor PPAR gamma. EPSs bind to TLR2 at the cell membrane, initiating downstream signaling via the adaptor protein MyD88, which activates both the IKK complex and MAPK pathways. This leads to the nuclear translocation of NF-κB and AP-1, transcription factors involved in inflammatory responses, which are subsequently suppressed by PPARγ activation. EPSs also influence fatty acid metabolism, increasing intracellular fatty acid levels, which, in turn, further modulate the PPARγ activity. Activated PPARγ, in complex with RXR and regulated via SUMOylation, inhibits inflammatory gene expression and promotes the transcription of genes involved in lipid metabolism, including the upregulation of ATGL, HSL, and CPT1 and the downregulation of Fabp4, aP2, and FAS. Additionally, EPSs enhance glucose uptake through increased Glut2 expression and the activation of glucokinase (GK) while promoting the activation of the AMPK and MAPK pathways, which leads to decreased glucose-6-phosphatase (G6P) activity. EPSs also upregulate SIRT1, which contributes to metabolic regulation, and modulate cholesterol and fatty acid metabolism through the activation of Cyp7a1 and Cd36, as well as the inhibition of HMG-CoA reductase.

**Table 1 microorganisms-13-01522-t001:** Key features of various EPSs that potentially contribute to cardiovascular health and lipid metabolism.

Function	Proposed Mechanism	Microorganisms	Study Type	References
Antioxidant activity	Free radical scavenging. Activation via the Keap1-Nrf2/ARE pathway. Protective effect against DNA damage and H_2_O_2_-induced oxidative stress.	*Lacticaseibacillus rhamnosus ACS5* *Limosilactobacillus fermentum A10* *Bifidobacterium* spp. *Leuconostoc* spp. *Weissella* spp. *Enterococcus* spp. *Lactococcus* spp.	In vitro	[92,93,94,95]
Reduction in cholesterol	Cholesterol adsorption and precipitation with bile salts. Decrease in plasma and serum lipid profile. Deconjugation of bile salts.	*Kefir (Lactobacillus kefiranofaciens)* *Lactiplantibacillus paraplantarum* NCCP *962* *Limosilactobacillus fermentum* NCDC400 *Schleiferilactobacillus harbinensis Z171* *Enterococcus faecium* F12	In vitro clinical trial	[96,97,98,99]
Immunomodulation	Inhibition of cytokines IL-6 and TNF-α, and activation of IL-10. Improvement in macrophage viability and phagocytosis. Inhibition of the mitogen-activated protein kinase (MAPK) pathway linking to the nuclear factor kappa B (NF-κB) gene. Immunomodulation via Toll-like receptors (TLR2-TLR4).	*Bacillus licheniformis* BioE-BL11 *Leuconostoc mesenteroides* BioE-LMD18, 201607 *Enterococcus hirae* WEHI01 *Lactiplantibacillus plantarum* DMDL 9010	In vitro, cellular model and animal models	[100,101,102,103,104]
Production of SCFAs	Fermentation for the production of Docosahexaenoic Acid (DHA). Production of fatty acid methyl esters (biodiesel).	*Schizochytrium* sp *Scenedesmus abundans*	In vitro with bioreactor	[105,106]
Angiogenesis	Improved HUVEC proliferation. Increased density of tubular structures. Promoted formation of growth factors. Increased phosphorylation of ERK kinase, c-Jun N-terminal kinase (JNK), and p38. Upregulated the expression of p21 and intercellular adhesion molecule 1 (ICAM1), as well as STAT3. Induced angiogenesis via HDAC5.	*Alteromonas infernus* *Neungee mushroom* Yeast beta-glucan	In vitro, cellular, animal model	[37,40,107]
Tissue regeneration	Increased cell adhesion. Maintenance of a confluent cell monolayer for 14 days. Stimulation of re-epithelialization and cell proliferation. Expression of endothelial markers (CD31 and vWf).	Pullulan/dextran EPS nanofibers Hydrogel containing (1→6)-β-D-glucan EPS from *Lasiodiplodia theobromae* MMPI	In vitro, cellular, animal model	[45,47]
Lower blood pressure	Inhibition of ECA	*Lactobacillus kefiranofaciens* *Streptococcus thermophiles* *Lactobacillus bulgaricus* *Streptococcus thermophilus*	Animal model, dietary supplementation	[77,84,108]

## Data Availability

No new data were created or analyzed in this study.

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
