# Peer review of "The Role of Microbial Exopolysaccharides in Preventing and Treating Cardiovascular Diseases"

_microorganisms, 2025, doi:10.3390/microorganisms13071522_

Round 1
Reviewer 1 Report
Comments and Suggestions for Authors
Recommendations:
- Please add in the introduction a search protocol for this study, briefly.
- Talk about the microbiome modulation and the impact on EPS, with probiotics/ fecal microbiota transplant.
- The figures are nice, but some tables with the main features that EPS influence in sections like Cardiovascular reconstruction/ lipid metabolism should be added.
Author Response
Reviewer #1
Comments and Suggestions for Authors
Recommendations:
- Please add in the introduction a search protocol for this study, briefly.
We greatly appreciate the time and effort of the reviewers in sharing their valuable comments with us.
Answer: Thank you for your recommendation. We added the following to the introduction:
A literature search was conducted in PubMed, Scopus, and Web of Science, covering the period from 1990 to 2024. The search strategy included the terms: "exopolysaccharides" OR "microbial polysaccharides" AND ("bacteria" OR "fungi" OR "archaea" OR "algae") AND ("cardiovascular diseases" OR "atherosclerosis" OR "lipid metabolism") AND ("carbohydrates" OR "biopolymers") AND ("probiotics" OR "microbiome modulation") AND ("angiogenesis" OR "antioxidant" OR "cholesterol-lowering" OR "tissue regeneration" OR "antihypertensive" OR "anticoagulant"). Peer-reviewed articles published in English between 1990 and 2024 were considered. Eligible studies included experimental, clinical, and review articles focusing on the role of microbial exopolysaccharides in the prevention or treatment of cardiovascular diseases, particularly regarding their angiogenic, antioxidant, cholesterol-lowering, tissue-regenerative, antihypertensive, and anticoagulant properties.
2. Talk about the microbiome modulation and the impact on EPS, with probiotics/ fecal microbiota transplant.
Answer: Thank you for your recommendation. We added the following to the section on General Characteristics of Exopolysaccharides:
The modulation of the gut microbiome has a significant influence on the production, structure, and functionality of microbial EPSs. For example, it has been shown that the intake of prebiotics such as inulin and fructooligosaccharides selectively promotes the growth of beneficial bacteria like Bifidobacteria and Lactobacilli, creating a favorable environment for the synthesis of complex and bioactive EPSs by the enteric microbiota, including immunomodulatory and anti-inflammatory properties. In addition, in vitro studies have demonstrated that EPSs themselves can act as microbiota modulators by increasing the production of short-chain fatty acids (SCFAs), organic acids, and other biomolecules that support the proliferation of beneficial bacteria, thereby contributing to the modulation of bacterial diversity in the gut and human fecal cultures. Likewise, factors such as diet, antibiotic use, or fecal microbiota transplantation can significantly alter the structure of the intestinal ecosystem, leading to the reprogramming of microbiota diversity and, consequently, the various microbial metabolic pathways involved in EPS biosynthesis.
3. The figures are nice, but some tables with the main features that EPS influence in sections like Cardiovascular reconstruction/ lipid metabolism should be added.
Answer: Thank you for your comment. We have added a table with the suggested information (pages 10-11).

Reviewer 2 Report
Comments and Suggestions for Authors
This review comprehensively addresses the emerging role of microbial exopolysaccharides (EPS) in the prevention and treatment of cardiovascular diseases (CVDs). The manuscript is well-structured, covering various beneficial properties of EPS, including antioxidant, antihypertensive, regenerative, anticoagulant, and cholesterol-reducing activities. The inclusion of detailed molecular mechanisms and signaling pathways significantly strengthens the review, making it a valuable resource for researchers in the field.
Abstract
Clarity on "pandemics" (Line 19): While CVDs are a global health crisis, referring to them as "one of the pandemics" might be slightly imprecise in a strict epidemiological sense, as the term usually refers to infectious diseases. Consider rephrasing to "one of the major global health crises" or "a leading cause of mortality and disability worldwide."
Conciseness of EPS Properties (Lines 24-28): The list of properties is good, but "updating their regenerative role in vascular epithelia, thus classifying EPS as biomolecules with great potential in the prevention of CVDs" could be slightly condensed. Perhaps, "highlighting their roles as antioxidants, antihypertensives, cholesterol regulators, and their regenerative effects on vascular epithelia, positioning EPS as promising biomolecules for CVD prevention."
Introduction
CVD Definition and Scope (Lines 34-39): The introduction of CVDs is clear. Ensure consistency in listing specific CVDs throughout the paper if you choose to detail them (e.g., if you mention "cerebrovascular disease," ensure "stroke" is also covered, which it is, but just a check for future consistency).
Flow of Pathophysiology (Lines 54-64): The explanation of oxidative and nitrosative stress in metabolic syndrome and their link to atherosclerosis and cardiac damage is well done. The flow between these concepts is generally good.
Strength of "multiple long-term side effects" (Lines 70-75): While valid, the current phrasing "in many cases, long-term side effects have been related to the current treatments" could be strengthened. You provide good examples; perhaps a stronger introductory phrase like "Despite the availability of effective pharmacological treatments, their widespread use is often limited by significant long-term side effects."
General Characteristics of Exopolysaccharides
Figure 1 Clarity (Lines 96-103): The figure is informative. The caption thoroughly explains the figure. Ensure that all elements in the figure (e.g., different types of bonds) are clearly distinguishable if color or symbol differentiation is key.
Biotechnological Applications (Lines 108-112): This section mentions that EPS properties "can be exploited by biotechnological applications." While good, you could briefly hint at how these properties lead to such applications within the review's scope (e.g., "for the synthesis of specific EPS with particular biomedical properties").
Overlapping References (Lines 113-114 and 117-118): Some references appear multiple times in quick succession (e.g., [16,12-25] and [19-34] for properties; [16,17,18,19] for tissue engineering). Consolidate where possible to avoid redundant citations and improve readability.
Antioxidant Activity of EPSs
Mechanism of Sulfated Groups (Lines 135-139): The explanation that "sulfate groups in an EPS molecule can determine various conformational structures, which in turn increase antioxidant activity" is valuable. You could briefly elaborate on how these conformational changes enhance antioxidant activity (e.g., by exposing more reactive sites, improving accessibility for radical scavenging).
"Inefficient activity" (Line 143): "The glycosidic bonds of its EPSs promote an inefficient activity of donating electron or hydrogen atoms to form hydroxyl radicals" sounds contradictory to antioxidant properties. Re-read and clarify this sentence. It should likely state that these bonds do efficiently donate, or that a different type of bond is inefficient, or that this particular strain's EPS reduces the ability to form hydroxyl radicals.
NRF2/KEAP1 Pathway Explanation (Lines 162-171): The explanation of the NRF2/KEAP1 pathway is detailed and clear. This level of detail is excellent for a review.
Figure 2 (Lines 173-188): The figure is very clear and visually supports the text. Ensure consistency between the figure and text, particularly on enzyme names (e.g., GSH, CAT, SOD are consistently used).
Antihypertensive Activity of EPS
ACE Inhibition Mechanism (Lines 228-229): "This enzyme is involved in the catalysis of angiotensin I to angiotensin II, leading to a significant increase in vasoconstriction." This is correct. You could briefly add that ACE also degrades bradykinin, a vasodilator, further contributing to its role in blood pressure regulation, and that ACE inhibitors prevent both actions.
Molecular Weight vs. Functionality (Lines 262-264): The statement that "structural complexity and high molecular weight of EPSs may not necessarily play a favorable role in ACE inhibition" is a key insight. Elaborate briefly on why specific structural motifs might be more important than overall size.
Cardiovascular Regeneration properties of Exopolysaccharides
Irreversible Damage (Line 275): "It is essential to consider that in myocardial infarction, the damage to cardiomyocytes can be irreversible due to the processes of ischemia and necrosis that occur." This is a strong statement. Reiterate how EPS mitigate this irreversibility or promote regeneration despite it.
HE800 DROS (Lines 323-328): The explanation of HE800 DROS and its mechanism is good. Ensure it's clear how inhibiting matrix metalloproteinases (MMPs) specifically promotes tissue regeneration in the context of cardiovascular health.
Figure 3 (Lines 380-396): This figure effectively illustrates the complex signaling pathways. Ensure the flow of arrows and labels is intuitive and easy to follow.
Anticoagulant Activity of Exopolysaccharides
Figure 4 (Lines 490-497): This figure on the coagulation cascade is crucial and well-represented. Ensure all components mentioned in the text (e.g., XIIa, XIa, IXa, Xa, thrombin, antithrombin III, HCII) are clearly labeled and their interactions accurately depicted.
Heparin vs. Sulfated EPS (Lines 449-450): "similar to heparin, but with a significantly longer effect." Quantifying or explaining why the effect is longer (e.g., slower degradation, different binding kinetics) would enhance this point.
EPSs Improve Lipid Metabolism and Cholesterol Reduction
Pancreatic Lipases (Lines 505-507): "This is because ionic charges and sulfate groups of EPSs interact with the amino groups of lipase enzymes, inactivating them." This is a good mechanistic detail.
Figure 5 (Lines 601-615): The figure provides a detailed overview of the signaling pathways related to cholesterol reduction. Ensure the legend accurately and completely describes all elements and arrows.
Specific Genes/Enzymes (Lines 590-596): You list many genes and enzymes (ATGL, HSL, CPT1, Fabp4, aP2, FAS, GLUT2, GK, SIRT1, G6P, Cyp7a1, Cd36, HMG-CoA reductase). While important, consider if a brief parenthetical description of their role (e.g., "ATGL, HSL, and CPT1 (involved in lipid breakdown)") would enhance readability for a broader audience without over-complicating.
Conclusions
Future Research (Lines 632-633): "If future research in this field progresses, EPSs could become a crucial tool in cardiovascular medicine." This is a good concluding thought. You might also consider adding a sentence about the challenges that need to be overcome for EPS to be widely adopted (e.g., standardization, production scalability, clinical trials).
Minor Edits/Typos
Line 1: The page number "3" seems to be part of the header. Ensure it's consistently excluded from the main text when referring to lines.
Line 10-11: "enriquesanhueza@comunidad.una" seems incomplete. Should be "enriquesanhueza@comunidad.unam.mx".
Line 90: "EPSs typically present structural arrangements, including hydroxyl groups, carbon-free radicals, mannose monomers, glucose, fucose, uronic acids, sulfate groups, branches, functional groups, and the presence of ẞ and a-type glucosidic bonds [25-35]." Remove "carbon-free radicals" from this list, as this is a reactive species, not a structural component of EPS.
Line 99: "β−(2→6)" and other bond types in the caption, the arrows are correct in LaTeX.
Line 117: "tissue regeneration essays and treatments" should probably be "tissue regeneration assays and treatments". "Essays" typically refers to written works, while "assays" refers to experimental tests.
Line 142: "Lysinibacillus sphaericus Ya6 strain demonstrated that the glycosidic bonds of its EPSs promote an inefficient activity of donating electron or hydrogen atoms to form hydroxyl radicals" - Re-evaluate this sentence for clarity and accuracy as noted above.
Line 149: "Fρ2+ or CH2+" should be Fe2+ or Cu2+.
Line 160: "Nuclear factor erythroid 2-related factor 2/Kelch-like ECH-associated protein 1 (NRF2/KEAP1)" - check capitalization for KEAP1 (sometimes all caps, sometimes Keap1). Be consistent.
Line 166: "RING-box protein 1 (RBX1) and Cullin 3 (Cul3)" - ensure consistency in Cul3 vs. CUL3.
Line 204: "depressed QRS waves" could be "attenuated QRS waves" or "reduced QRS amplitude" for more scientific precision.
Line 231-232: "5 mg/mL" - ensure consistent use of LaTeX for units throughout the document (which it mostly is, good job).
Line 287: "Histone deacetylase 5 (HDAC5)" - check capitalization for HDAC5 (sometimes all caps, sometimes HDAc5). Be consistent.
Line 290: "serine/threonine-type protein kinase (Akt) and Erk1/2 factors" - consistency for ERK1/2 vs. Erk1/2.
Line 352: "42 μm" - ensure consistent LaTeX for mu.
Line 414: "ERK 1/2" - ensure consistent LaTeX for 1/2.
Line 439: "Figure 3" is incorrect, should be Figure 4.
Line 599: "Th17 cells" - check consistency on Th17 vs. TH17.
Author Response
Reviewer #2
We greatly appreciate the time and effort of the reviewers in sharing their valuable comments with us.
Comments and Suggestions for Authors
This review comprehensively addresses the emerging role of microbial exopolysaccharides (EPS) in the prevention and treatment of cardiovascular diseases (CVDs). The manuscript is well-structured, covering various beneficial properties of EPS, including antioxidant, antihypertensive, regenerative, anticoagulant, and cholesterol-reducing activities. The inclusion of detailed molecular mechanisms and signaling pathways significantly strengthens the review, making it a valuable resource for researchers in the field.
Abstract
Clarity on "pandemics" (Line 19): While CVDs are a global health crisis, referring to them as "one of the pandemics" might be slightly imprecise in a strict epidemiological sense, as the term usually refers to infectious diseases. Consider rephrasing to "one of the major global health crises" or "a leading cause of mortality and disability worldwide."
Answer: Agree. The term “pandemics” was changed to “one of the major global health crises.”
Conciseness of EPS Properties (Lines 24-28): The list of properties is good, but "updating their regenerative role in vascular epithelia, thus classifying EPS as biomolecules with great potential in the prevention of CVDs" could be slightly condensed. Perhaps, "highlighting their roles as antioxidants, antihypertensives, cholesterol regulators, and their regenerative effects on vascular epithelia, positioning EPS as promising biomolecules for CVD prevention."
Answer: Agree. Done.
Introduction
CVD Definition and Scope (Lines 34-39): The introduction of CVDs is clear. Ensure consistency in listing specific CVDs throughout the paper if you choose to detail them (e.g., if you mention "cerebrovascular disease," ensure "stroke" is also covered, which it is, but just a check for future consistency).
Answer: Thank you for your detailed comment. We have added related information on coronary artery disease, peripheral artery disease, venous thrombosis, and pulmonary embolism to the sections “EPSs Improve Lipid Metabolism and Cholesterol Reduction”, “Antihypertensive Activity of EPS”, and “Anticoagulant Activity of Exopolysaccharides” to ensure consistent coverage of these topics.
Flow of Pathophysiology (Lines 54-64): The explanation of oxidative and nitrosative stress in metabolic syndrome and their link to atherosclerosis and cardiac damage is well done. The flow between these concepts is generally good.
Answer: Thank you for your comment.
Strength of "multiple long-term side effects" (Lines 70-75): While valid, the current phrasing "in many cases, long-term side effects have been related to the current treatments" could be strengthened. You provide good examples; perhaps a stronger introductory phrase like "Despite the availability of effective pharmacological treatments, their widespread use is often limited by significant long-term side effects."
Answer: Agree. The introductory phrase has been replaced with the one you recommended.
General Characteristics of Exopolysaccharides
Figure 1 Clarity (Lines 96-103): The figure is informative. The caption thoroughly explains the figure. Ensure that all elements in the figure (e.g., different types of bonds) are clearly distinguishable if color or symbol differentiation is key.
Answer: Thank you for your comment. We have ensured that all the elements in the figure are clearly distinguishable and enhanced them. The figure was constructed based on the international monosaccharide symbol nomenclature (Symbol Nomenclature for Glycans). Besides, we provided a little more information in the figure caption.
Biotechnological Applications (Lines 108-112): This section mentions that EPS properties "can be exploited by biotechnological applications." While good, you could briefly hint at how these properties lead to such applications within the review's scope (e.g., "for the synthesis of specific EPS with particular biomedical properties").
Answer: Thank you for your recommendation. We added the following idea:
e.g., for the synthesis of specific EPS with particular or enhanced biomedical properties.
Overlapping References (Lines 113-114 and 117-118): Some references appear multiple times in quick succession (e.g., [16,12-25] and [19-34] for properties; [16,17,18,19] for tissue engineering). Consolidate where possible to avoid redundant citations and improve readability.
Answer: Thank you for your recommendation. We eliminated redundant references.
Antioxidant Activity of EPSs
Mechanism of Sulfated Groups (Lines 135-139): The explanation that "sulfate groups in an EPS molecule can determine various conformational structures, which in turn increase antioxidant activity" is valuable. You could briefly elaborate on how these conformational changes enhance antioxidant activity (e.g., by exposing more reactive sites, improving accessibility for radical scavenging).
Answer: Thank you for your comment. We added the following ideas:
Such conformational changes, resulting from the electrostatic repulsion between negatively charged sulfate groups, can enhance the hydration of the EPS, making it more soluble and able to interact with aqueous solutions. This thereby exposes more reactive sites, improving accessibility for radical scavenging, as well as directly capturing ROS generated during the lipid peroxidation chain.
"Inefficient activity" (Line 143): "The glycosidic bonds of its EPSs promote an inefficient activity of donating electron or hydrogen atoms to form hydroxyl radicals" sounds contradictory to antioxidant properties. Re-read and clarify this sentence. It should likely state that these bonds do efficiently donate, or that a different type of bond is inefficient, or that this particular strain's EPS reduces the ability to form hydroxyl radicals.
Answer: Thank you for your comment. You are right; we had a written mistake. The sentence was corrected as follows:
The glycosidic bonds in its EPS facilitate efficient electron or hydrogen atom transfer, leading to the scavenging of hydroxyl radicals.
NRF2/KEAP1 Pathway Explanation (Lines 162-171): The explanation of the NRF2/KEAP1 pathway is detailed and clear. This level of detail is excellent for a review.
Answer: Thank you for your comment.
Figure 2 (Lines 173-188): The figure is very clear and visually supports the text. Ensure consistency between the figure and text, particularly on enzyme names (e.g., GSH, CAT, SOD are consistently used).
Answer: Done.
Antihypertensive Activity of EPS
ACE Inhibition Mechanism (Lines 228-229): "This enzyme is involved in the catalysis of angiotensin I to angiotensin II, leading to a significant increase in vasoconstriction." This is correct. You could briefly add that ACE also degrades bradykinin, a vasodilator, further contributing to its role in blood pressure regulation, and that ACE inhibitors prevent both actions.
Answer: Thank you for your recommendation. We rewrote the sentences as follows:
This enzyme is involved in two key processes that increase vasoconstriction in the system. First, ACE promotes the catalysis of angiotensin I to angiotensin II. Additionally, it promotes the degradation of the vasodilator bradykinin, resulting in a significant increase in vasoconstriction.
Molecular Weight vs. Functionality (Lines 262-264): The statement that "structural complexity and high molecular weight of EPSs may not necessarily play a favorable role in ACE inhibition" is a key insight. Elaborate briefly on why specific structural motifs might be more important than overall size.
Answer: Thank you for your recommendation. We added the following information:
These findings suggest that the structural complexity and high molecular weight of EPSs may not necessarily play a favorable role in ACE inhibition, indicating that EPS functionality primarily depends on specific structural motifs and even culture conditions. For instance, EPSs exhibiting benzoylation and acetylation motifs, such as those synthesized by the Lachnum p. YM130 strain (as previously explained), have demonstrated significant cardioprotective effects. The potential role of EPS acetylation in counteracting hypertension can be elucidated by comparing it with the acetylation mechanisms observed in other pharmacological agents. For example, aspirin's acetylation ability allows it to irreversibly inactivate enzymes such as Cyclooxygenase-1 (COX-1) and Cyclooxygenase-2 (COX-2). While primarily recognized for its antiplatelet and anti-inflammatory properties, aspirin, particularly at low doses, has been shown to modestly influence blood pressure by affecting the synthesis of prostaglandins and thromboxanes, which play crucial roles in regulating vascular tone.
Cardiovascular Regeneration properties of Exopolysaccharides
Irreversible Damage (Line 275): "It is essential to consider that in myocardial infarction, the damage to cardiomyocytes can be irreversible due to the processes of ischemia and necrosis that occur." This is a strong statement. Reiterate how EPS mitigate this irreversibility or promote regeneration despite it.
Answer: Thank you for your recommendation. We added the following paragraph:
On this line, EPSs have shown myocardial-regenerative effects, even amidst the chronic damage provoked by CVDs. EPSs mitigate this damage preventively by promoting several differentiation, regenerative, cell migration, and neovascularization signaling pathways, thereby favoring the differentiation and migration of resident cells towards damaged areas, which can contribute to partial functional regeneration of myocardial tissue.
HE800 DROS (Lines 323-328): The explanation of HE800 DROS and its mechanism is good. Ensure it's clear how inhibiting matrix metalloproteinases (MMPs) specifically promotes tissue regeneration in the context of cardiovascular health.
Answer: Thank you for your recommendation. We added the following paragraph:
In the context of cardiovascular health, this inhibition is essential as it helps minimize structural damage to the heart and blood vessels during inflammatory or ischemic events. A more supportive environment for tissue regeneration is established by maintaining the integrity of the extracellular matrix. Inhibiting matrix metalloproteinases (MMPs) promotes cellular repair, revascularization, and the functional restoration of the affected tissues.
Figure 3 (Lines 380-396): This figure effectively illustrates the complex signaling pathways. Ensure the flow of arrows and labels is intuitive and easy to follow.
Answer: Thank you for your comment. We revisited the figure to ensure that the flow of arrows and labels was intuitive and easy to follow.
Anticoagulant Activity of Exopolysaccharides
Figure 4 (Lines 490-497): This figure on the coagulation cascade is crucial and well-represented. Ensure all components mentioned in the text (e.g., XIIa, XIa, IXa, Xa, thrombin, antithrombin III, HCII) are clearly labeled and their interactions accurately depicted.
Answer: Thank you for your comment. We revisited the figure to ensure that the coagulation cascade and its components are accurately presented.
Heparin vs. Sulfated EPS (Lines 449-450): "similar to heparin, but with a significantly longer effect." Quantifying or explaining why the effect is longer (e.g., slower degradation, different binding kinetics) would enhance this point.
Answer: Thank you for your comment. We added the following paragraph:
This can occur by delaying the thromboplastin-activated thrombin generation, resulting in a significantly longer lag phase of the contact-induced thrombin generation system compared to that of heparin and fucoidan, as demonstrated for various EPS derivatives.
EPSs Improve Lipid Metabolism and Cholesterol Reduction
Pancreatic Lipases (Lines 505-507): "This is because ionic charges and sulfate groups of EPSs interact with the amino groups of lipase enzymes, inactivating them." This is a good mechanistic detail.
Answer: Thank you for your comment.
Figure 5 (Lines 601-615): The figure provides a detailed overview of the signaling pathways related to cholesterol reduction. Ensure the legend accurately and completely describes all elements and arrows.
Answer: Thank you for your comment. We revisited the figure to ensure that the legend wholly and accurately describes all elements and arrows.
Specific Genes/Enzymes (Lines 590-596): You list many genes and enzymes (ATGL, HSL, CPT1, Fabp4, aP2, FAS, GLUT2, GK, SIRT1, G6P, Cyp7a1, Cd36, HMG-CoA reductase). While important, consider if a brief parenthetical description of their role (e.g., "ATGL, HSL, and CPT1 (involved in lipid breakdown)") would enhance readability for a broader audience without over-complicating.
Answer: Thank you for your recommendation. We added the following parenthetical descriptions:
ATGL, HSL, and CPT1 (involved in lipid degradation); Fabp4 y aP2 (involved in intracellular transport of fatty acids); FAS (involved in fatty acid synthesis); GLUT2 and GK (both involved in glucose uptake and phosphorylation); SIRT1 (involved in energy regulation and cellular longevity); G6P (involved in gluconeogenesis); Cyp7a1 (involved in bile acid synthesis); Cd36 (involved in fatty acid uptake); and HMG-CoA reductase (involved in cholesterol synthesis).
Conclusions
Future Research (Lines 632-633): "If future research in this field progresses, EPSs could become a crucial tool in cardiovascular medicine." This is a good concluding thought. You might also consider adding a sentence about the challenges that need to be overcome for EPS to be widely adopted (e.g., standardization, production scalability, clinical trials).
Answer: Thank you for your recommendation. We added the following ideas:
Nonetheless, to achieve this goal, challenges related to standardizing methods for producing high quantities of pure and homogeneous EPS structures must be overcome. Such difficulties arise because the EPS structure and function are highly dependent on the culture conditions and the characteristics of the microorganism strain used for their production. Furthermore, clinical trials to demonstrate the efficacy of using exopolysaccharides for treating these kinds of pathologies are lacking. Then, a greater effort is required to convince healthcare decision-makers to consider exopolysaccharides as a suitable alternative for treating conditions such as CVDs.
Minor Edits/Typos
Line 1: The page number "3" seems to be part of the header. Ensure it's consistently excluded from the main text when referring to lines.
Answer: Thank you.
Line 10-11: "enriquesanhueza@comunidad.una" seems incomplete. Should be "enriquesanhueza@comunidad.unam.mx".
Answer: Thank you. We completed it.
Line 90: "EPSs typically present structural arrangements, including hydroxyl groups, carbon-free radicals, mannose monomers, glucose, fucose, uronic acids, sulfate groups, branches, functional groups, and the presence of ẞ and a-type glucosidic bonds [25-35]." Remove "carbon-free radicals" from this list, as this is a reactive species, not a structural component of EPS.
Answer: Done
Line 99: "β−(2→6)" and other bond types in the caption, the arrows are correct in LaTeX.
Answer: Yes, they are correct.
Line 117: "tissue regeneration essays and treatments" should probably be "tissue regeneration assays and treatments". "Essays" typically refers to written works, while "assays" refers to experimental tests.
Answer: Agree. Done.
Line 142: "Lysinibacillus sphaericus Ya6 strain demonstrated that the glycosidic bonds of its EPSs promote an inefficient activity of donating electron or hydrogen atoms to form hydroxyl radicals" - Re-evaluate this sentence for clarity and accuracy as noted above.
Answer: Agree. Done.
Line 149: "Fρ2+ or CH2+" should be Fe2+ or Cu2+.
Answer: Thank you for your comment. We corrected it.
Line 160: "Nuclear factor erythroid 2-related factor 2/Kelch-like ECH-associated protein 1 (NRF2/KEAP1)" - check capitalization for KEAP1 (sometimes all caps, sometimes Keap1). Be consistent.
Answer: Thank you for your comment. We normalized the writing of KEAP1 in the corrected manuscript.
Line 166: "RING-box protein 1 (RBX1) and Cullin 3 (Cul3)" - ensure consistency in Cul3 vs. CUL3.
Answer: Done.
Line 204: "depressed QRS waves" could be "attenuated QRS waves" or "reduced QRS amplitude" for more scientific precision.
Answer: Done.
Line 231-232: "5 mg/mL" - ensure consistent use of LaTeX for units throughout the document (which it mostly is, good job).
Answer: Done.
Line 287: "Histone deacetylase 5 (HDAC5)" - check capitalization for HDAC5 (sometimes all caps, sometimes HDAc5). Be consistent.
Answer: Done.
Line 290: "serine/threonine-type protein kinase (Akt) and Erk1/2 factors" - consistency for ERK1/2 vs. Erk1/2.
Answer: Done.
Line 352: "42 μm" - ensure consistent LaTeX for mu.
Answer: Done.
Line 414: "ERK 1/2" - ensure consistent LaTeX for 1/2.
Answer: Done.
Line 439: "Figure 3" is incorrect, should be Figure 4.
Answer: Done.
Line 599: "Th17 cells" - check consistency on Th17 vs. TH17.
Answer: Done.
